# Chromosome Number and Genome Size Evolution in *Brasolia* and *Sobralia* (Sobralieae, Orchidaceae)

**DOI:** 10.3390/ijms23073948

**Published:** 2022-04-01

**Authors:** Przemysław Baranow, Joanna Rojek, Magdalena Dudek, Dariusz Szlachetko, Jerzy Bohdanowicz, Małgorzata Kapusta, Iwona Jedrzejczyk, Monika Rewers, Ana Paula Moraes

**Affiliations:** 1Department of Plant Taxonomy & Nature Conservation, Faculty of Biology, University of Gdansk, Wita Stwosza 59, 80-308 Gdansk, Poland; przemyslaw.baranow@ug.edu.pl (P.B.); magdalena.dudek@ug.edu.pl (M.D.); dariusz.szlachetko@ug.edu.pl (D.S.); 2Department of Plant Cytology and Embryology, Faculty of Biology, University of Gdansk, Wita Stwosza 59, 80-308 Gdansk, Poland; jerzy.bohdanowicz@ug.edu.pl (J.B.); malgorzata.kapusta@ug.edu.pl (M.K.); 3Laboratory of Molecular Biology and Cytometry, Department of Agricultural Biotechnology, Bydgoszcz University of Science and Technology, Kaliskiego Ave 7, 85-796 Bydgoszcz, Poland; jedrzej@pbs.edu.pl (I.J.); mrewers@pbs.edu.pl (M.R.); 4Laboratory of Cytogenomic and Evolution of Plants, Center of Natural and Human Science, Federal University of ABC (UFABC), Sao Bernardo do Campo 09606-045, SP, Brazil; ana.moraes@ufabc.edu.br

**Keywords:** karyotype, phylogeny, nuclear DNA content, flow cytometry, dysploidy, phylogenomics, plant genetic diversity

## Abstract

Despite the clear circumscription of tribe Sobralieae (Orchidaceae), its internal relationships are still dubious. The recently delimited genus *Brasolia*, based on previous *Sobralia* species, is now assumed to be paraphyletic, with a third genus, *Elleanthus*, nested in it. The morphology of these three genera is significantly different, indicating the necessity of new data for a better genera delimitation. Though morphology and molecular data are available, cytogenetics data for Sobralieae is restricted to two *Sobralia* and one *Elleanthus* species. Aiming to evaluate the potential of cytogenetic data for *Brasolia-Elleanthus-Sobralia* genera delimitation, we present chromosome number and genome size data for 21 and 20 species, respectively, and used such data to infer the pattern of karyotype evolution in these genera. The analysis allowed us to infer x = 24 as the base chromosome number and genome size of average 1C-value of 5.0 pg for the common ancestor of *Brasolia-Elleanthus-Sobralia*. The recurrent descending dysploidy in Sobralieae and the punctual genome upsize suggest a recent diversification in Sobralieae but did not allow differing between *Brasolia* and *Sobralia*. However, the basal position of tribe Sobralieae in the subfamily Epidendroideae makes this tribe of interest to further studies clarifying the internal delimitation and pattern of karyotype evolution.

## 1. Introduction

Since Ruiz and Pavón established the genus *Sobralia* (Orchidaceae) in 1794 [1], its species reached about 200 taxa distributed from southern Mexico to Bolivia and Brazil, with a remarkable species diversity in the Andes. The species are often terrestrial, rarely epiphytic, usually tall, or very tall (for example, stems of *Sobralia luerorum* Dodson can reach up to 3.5 m height), with slender, cane-like stems growing in clumps and strongly developed root systems. The inflorescence position and structure have served as a basis for the infrageneric classification of *Sobralia* [2], with five recognized sections [2,3]. 

The nominal section was characterized by lateral or rarely terminal inflorescences with branching, well-developed raceme and relatively small floral bracts compared to the size of the ovary [2]. The section *Racemosae* Brieger, despite terminal inflorescences, could differ from the former by its elongated and unbranched inflorescences with large floral bracts. Section *Globosae* Brieger is composed of small plants with narrow leaf blades, small flowers positioned in terminal and condensed inflorescences (shortened internodes hidden under the floral bracts) that successively produce a single flower at a time and elongate with successively produced floral bracts. Species from section *Abbreviatae* Brieger share terminal and condensed inflorescences with the previous section but, instead, present floral bracts forming a cone. The fifth section, *Intermediae* Brieger, was established for a single taxon—*Sobralia fragrans* Lindl.—to emphasize its elongated basal internode of the inflorescences. Dressler [3] enlarged this section, placing other species with small flowers and inflorescences (Figure 1). 

Molecular phylogenetic studies [4,5] revealed that the nominal section of *Sobralia* is paraphyletic, indicating the necessity of taxonomic reassignment. Based on the morphological distinctness and the mentioned phylogenetic relations, the section *Sobralia* was elevated to the status of a separate genus named *Brasolia* (Rchb.f.) Baranow, Dudek and Szlach [4]. Dressler et al. [6] already had predicted the necessity of such reassignments, proposing to conserve the name *Sobralia* with *Sobralia biflora* Ruiz and Pav. as a nominal species to avoid transferring most of the genus representatives into *Cyathoglottis* Poepp. & Endl., the second-oldest generic name available for *Sobralia*. Thus, the former nominal section of the genus *Sobralia* now constitutes a separate genus renamed *Brasolia,* with the former section *Abbreviatae* being considered the *Sobralia* nominal section [4]. Nevertheless, the genus *Elleanthus* C. Presl was nested in the new *Brasolia* [4], an association previously suggested by Neubig et al. [5], suggesting further taxonomic realignments are necessary.

Despite the availability of molecular and morphological data, the cytogenetic information, which is known to significantly help to understand the relationship and delimitation among taxa [7], is limited in the tribe Sobralieae, with only three chromosome counts available until now: two *Sobralia* species, *S. liliastrum* Lindl. and *S. sessilis* Lindl., with 2*n* = 2x = 48, and one *Elleanthus*, *E. brasiliensis* (Lindl.) Rchb.f. with 2*n* = 2x = 50 [8]. Regarding the genome size, no data is available. The karyotype evolutionary history results from the successive accumulation of numerical and structural chromosome rearrangements (for a review, see [9,10]), making it mandatory to evaluate the cytogenetic data in a phylogenetic background. The family Orchidaceae Juss. is known for its large variation in both chromosome numbers, which varies more than 20-fold (from 2*n* = 10 to 2*n* = 240; [11,12]), and genome size, which varies more than 168-fold, the second-largest variability in Angiosperms [13]. It makes Orchidaceae an interesting clade for studies in karyotype evolution, and by inferring the historical karyotype and genome size modifications, it also gives clues about taxonomic delimitations [14,15]. However, the scare cytogenetic data in tribe Sobralieae prevents any further discussion on *Brasolia* and *Sobralia* delimitation, as well as the Sobralieae karyotype evolution.

Based on the statement that the more taxonomical data sources, the more realistic the classification becomes, we have chosen the karyotype evolution of Sobralieae as the next step to better understand the relationship between genera *Brasolia, Elleanthus* and *Sobralia*. Based on chromosome number and genome size for these three genera, we aim to answer the following questions: (1) Which are the main evolutionary mechanisms shaping karyotype and genome size in tribe Sobralieae? and (2) Based on the ancestral base chromosome number (x) and genome size (1C-value) for these three genera and tribe Sobralieae, is it possible to separate the genera? We gathered the somatic chromosome number for 21 species and the genome size for 20 species to answer these questions and interpreted all the data in a phylogenetic approach, based on Maximum Likelihood (ML) and Bayesian Inference (BI) analyses, both using nuclear ribosomal DNA (nrITS) and maturase K (*mat*K) regions.

## 2. Results

### 2.1. Phylogenetic Analysis Suggested Brasolia as Paraphyletic

Both applied methodologies (ML and BI) revealed the same general results, with *Sobralia* as a monophyletic genus and *Brasolia* as paraphyletic with *Elleanthus* nested in it (Figure 2 and Appendix A). Despite the general similarity between both reconstruction methods, some differences could be detected, mainly inside *Brasolia* regarding the position of *B. portillae* and *B. mandonii* in relationship to the remainder *Brasolia* species (Appendix A). While this group was placed as a sister of remainder *Brasolia* and *Elleanthus* in ML, it was placed as a sister of *B. flava* and *Elleanthus* in BI. Considering the slight differences between ML and BI, we used both phylogenies to further steps. Both trees were pruned, removing species without chromosome number and genome size data, saving the tree topology for following analysis (Figure 2).

### 2.2. Chromosome Number and Genome Size Varies in Sobralieae

Chromosome numbers (2*n*) are provided for six *Brasolia*, two *Elleanthus* (Figure 3 and Appendix A), and 21 *Sobralia* species (Figure 4 and Appendix A), with two additional reports from the literature, totalizing chromosome numbers to 23 Sobralieae species (Table 1). Chromosome numbers varied from 2*n* = 40 to 2*n* = 48 in *Brasolia* and *Sobralia*, while both *Elleanthus* species presented 2*n* = 50 (Table 1).

The DNA content was estimated for 20 species (six *Brasolia*, two *Elleanthus,* and 12 *Sobralia*), ranging from 1C = 2.04 pg in *E. crinipes* to 1C = 8.05 pg in S. *luerorum,* reflecting a four-fold difference in genome size values (Table 1). According to the categorization proposed by Soltis et al. [16], three species can be considered as having small genomes (1C-value varying from 2.04 to 3.43 pg) and 17 as having intermediate genome sizes (1C-value varying from 3.77–8.05 pg; Table 1). 

The genome size among *Brasolia* species ranged from 1C = 6.37 pg in *B. altissima* to 1C = 3.30 pg in *B. portillae*, representing a two-fold variation. According to 1C-value, *Brasolia* species could be organized into four groups (F = 3.719, *p* < 2 × 10^−16^; Figure 5a): the first represented by *B. altissima* with the largest genome size, the second group formed by *B. cattleya*-*B. dichotoma*-*B. flava*, the third composed by *B. hirtzii*, and the last group formed by *B. portillae* with the smallest genome size found in the genus (Figure 5a). The two *Elleanthus* species differ two-fold in their genome size, which enabled the identification of those two species based on the genome size (F = 5.622, *p* < 2 × 10^−16^; Table 1). Among the 12 *Sobralia* species, the estimated 1C-value presented a significant difference among species (F = 650, *p* < 2 × 10^−16^; Figure 5b), ranging from 1C = 3.43 pg in *S. fragrans* to 1C = 8.05 pg in *S*. *luerorum* (Table 1; Figure 5b). The three larger genome sizes were found in *S. luerorum*, *S. macrantha*, and *S. rosea*, while the two smaller ones were found in *S. crocea* and *S. frag**rans,* while the remaining species formed a heterogeneous group without clear differentiation among species (Figure 5b).

### 2.3. Ancestral Reconstruction of Chromosome Number and Genome Size Indicate x = 25 and Small Genome Size for the Common Ancestor of Brasolia-Elleanthus-Sobralia

Regarding the ancestral reconstruction for chromosome number, for both trees, the ChromEvol suggested the same model as the best one: constant rate without duplication (Table 2). However, considering the ΔAIC < 2, no differences were found among constant rate, constant rate with demi-polyploidy and the best model for ML and BI phylogenies (Table 2). The Model Adequacy test confirmed the null model, constant rate without polyploidy, as the most adequate considering the variance and entropy tests (*p* = 0.13 and 0.21 for ML and *p* = 0.31 and 0.35 for IB, respectively) but not the parsimony (*p* = 0.00 for both ML and BI), suggesting that the chromosome number transitions are variable among clades. The best models highlight the importance of dysploidy variation as the main mechanism in chromosome evolution in *Brasolia,*
*Elleanthus* and *Sobralia*. 

The best model applied in both phylogenetic trees revealed similar results for most phylogenetic nodes. The base chromosome number x = 24 was suggested for the tribe Sobralieae, as well as for the genera *Brasolia* and *Sobralia*, while x = 25 was suggested for genus *Elleanthus* (Figure 2). The descending dysploidy was more frequently detected than ascending dysploidy, while no duplication was observed (Appendix A). Successive descending dysploidies were observed in both phylogenetic trees reaching x = 23 in the large group comprehending *S. atropubescens*, *S. sessilis, S. yauaperyensis, S. decora, S. fimbriata*, *S. powellii* in IB phylogeny, but also grouping *S. bimaculata*, *S. vallecaucana*, *S. crocea*, and *S. fragrans* in ML phylogeny (Figure 2). Further descending dysploidies could be observed on *S. crocea* and *S. frag**rans**,* reaching x = 22, and *B. cattleya* and *B. dichotoma*, reaching x = 23 (ML and IB). Only one ascending dysploidy was detected in both trees, on the node of *Elleanthus* genus with x = 25 (Figure 2).

Despite statistics test could not differentiate *Brasolia* and *Sobralia* based on genome size (F = 0.488; *p* = 0.487), the ancestral reconstruction for genome size suggested a higher ancestral 1C-value for *Sobralia* (1C = 5.5 pg) compared to *Brasolia* (1C = 4.45–4.73 pg) (Figure 2). Some punctual amplifications could be observed inside each genus, even without any important chromosome number variation. For example, the species *S. wilsoniana* (1C = 4.25 pg), *S. rosea* (1C = 5.93 pg), *S. macrantha* (1C = 6.24 pg), and *S. luerorum* (1C = 8.05 pg) strongly vary among their 1C-values (Appendix A), although they present the same chromosome number 2n = 48 (*n* = 24; Figure 2). Inside *Brasolia*, genome size increase could be observed in *B. altissima* (1C = 6.37 pg, 2n = 46; Figure 2). A genome size reduction could be observed in *S. crocea* (1C = 3.77 pg, 2*n* = 44) and *S. frag**rans* (1C = 3.43 pg, 2*n* = 42) and in *B. portillae* (1C = 3.3 pg, 2*n* = 48). An additional important genome size reduction was also observed in *E. crinipes* that presented half of the genome size (1C = 2.04 pg) compared with *E. brasiliensis* (1C = 4.79 pg), but with the same chromosome number (2*n* = 50 for both species; Figure 2).

Chromosome number presents no phylogenetic signal considering both phylogenetic trees, ML and BI, and tested indexes, Pagel’s lambda and Blomberg’s K suggesting that phylogenetic history has a minimal effect on both traits (Table 3). Otherwise, genome size seems to present phylogenetic signal considering both phylogenetic trees, ML and BI, and tested indexes, Pagel’s lambda (values very close to 1) and Blomberg’s K (values equal or superior to 1; see Table 3). Following the concept of phylogenetic signal, i.e., related species seems to present similar traits more than they resemble species drawn at random in this group, the phylogenetic signals for genome size suggested that this trait could reflect the evolutionary history of *Brasolia-**Elleanthus-Sobralia*, being a potential informative trait to discuss the genera delimitation. 

## 3. Discussion

### 3.1. Phylogenetic Reconstruction

Both methodologies, ML and BI, presented the same general topology and agreed with previously published literature [4,5]. Baranow et al. [4] and Neubig et al. [5] noted that *Brasolia* species (along with *Elleanthus*) and *Sobralia* were placed on separate clades. It explicitly confirms the Dressler [6] attempt, followed by Baranow et al. [4], to place these two groups in two genera. The relationship between *Brasolia* and *Elleanthus* is hard to explain, especially considering the morphological distinctness between these two genera (compare Figure 1a–c with Figure 1d–j). The morphological differences between *Brasolia* and *Elleanthus* are much more visible than *Sobralia* and *Brasolia* morphological differences (compare Figure 1a–c with Figure 1d–j). Despite this, *Sobralia* and *Brasolia* formed two clades, both with high support. However, *Elleanthus,* even presenting apparent morphological differences from *Brasolia*, was placed nested on it in both phylogenies. Taking into account the present and previous phylogenetic studies, e.g., [4,5,6], one can suggest that *Brasolia* should be transferred to *Elleanthus*. However, from the practical point of view, it is not supported as the taxa are morphologically distinct*,* and the *Elleanthus* sampling is low. The *Elleanthus* taxonomic circumscription remains an open question for future studies in which a broader Sobralieae sampling should be mandatory.

### 3.2. Ancestral Base Chromosome Number Reconstruction Indicates Dysploidy as the Main Mechanism of Karyotype Evolution

There were just three available data regarding Sobralieae chromosome number [8] and no data about genome size. However, such data is now enlarged to 23 chromosome numbers and 20 genome sizes. The chromosome number reported to *Sobralia* [8], 2*n* = 48, was recurrent among species analyzed here, present in 38.1% of the counts. Furthermore, we confirm 2*n* = 50 to *Elleanthus brasiliensis* [8].

Our data showed a sizeable dysploid variation in both *Brasolia* and *Sobralia*, confirmed by the ancestral reconstruction analysis that suggested the descending dysploidy as the main mechanism of chromosome change in Sobralieae. Nevertheless, one should keep in mind that here dysploidy refers to chromosome number variation by one or few chromosomes and, in this sense, it includes both aneuploidy and dysploidy (for review see [9,17]). The importance of dysploidy in orchid chromosome evolution was already suggested in different orchid groups as subtribe Pleurothaliidinae (tribe Cymbidieae [18]) and the genera *Christensonella* Szlach., Mytnik, Górniak and Śmiszek [19], *Heterotaxis* Lindl. [16], *Bifrenaria* Lindl. and close genera [20], all from subtribe Maxillariinae, tribe Cymbidieae. In *Christensonella* and *Heterotaxis*, it was possible to map the point involved in the fusion-fission rearrangement causing the dysploid variation, confirming the occurrence of fission-fusion (i.e., dysploidy variation) associated with DAPI^+^ heterochromatic bands, reducing from 2*n* = 38 to 2*n* = 36 in the former genus [19] or with rDNA 45S, reducing from 2*n* = 42 to 2*n* = 40 in the last genus [15]. In tribe Sobralieae, we could confirm the importance of descending dysploidy with chromosome number reduction from 2*n* = 48 (*n* = 24) to 2*n* = 46, 44, 42 in *Sobralia* and *Brasolia* genera. However, based on the homogeneous chromosome morphology presented in the species analyzed here, it was impossible to infer the fission-fusion points or chromosome loss, causing the chromosome number reduction, making mandatory the use of in situ hybridization techniques.

The frequent occurrence of dysploidy in Sobralieae is in accordance with the diploidisation period following whole-genome duplication. Polyploidy is a central evolutionary mechanism in plants, but its occurrence is not random or constant throughout plant evolution [21,22,23]. Otherwise, it is suggested that polyploidy pulses occurred with high frequency at some geological periods as the Cretaceous-Paleogene transition (K-Pg; 66 Mya), Paleogene (Paleocene-Eocene transition; 56 Mya), and Neogene (at Late Miocene; 11 - 5 Mya), configuring periods of major stress (reviewed by [24]). Such concentration of polyploidy events in periods of geological history is confirmed by a high concentration of polyploid basal nodes in the Angiosperms phylogeny, representing some large extant plant families [23,24,25,26], which could be true for Orchidaceae with an estimated age of 90 million years ago (Mya; [27,28,29,30]). Furthermore, the suggested base chromosome number in Orchidaceae is x = 7 [8] or x = 10 [31], which suggests that most extant orchid species are polyploids. Such an assumption could be applied to the tribe Sobralieae (x = 24). 

The tribe Sobralieae is placed on the base of the subfamily Epidendroideae (44.50–60.27 Mya; [30]), the largest Orchidaceae subfamily [32]. Sobralieae possibly diverged at 45 Mya on the Neotropics, with *Brasolia*, *Elleanthus*, and *Sobralia* diverging more recently, at ca. 7–10 Mya [27,28,30]. Based on the previously assumed base chromosome number of Orchidaceae, the tribe Sobralieae (x = 24) should be considered a hexaploid followed by descending dysploidy (x = 7 -> 14 -> 28 -> 26 -> 24) or tetraploid followed by ascending dysploidy (x = 10 -> 20 -> 22 -> 24). The genomic redundancy generated by polyploidy were “accommodated” in a diploid genome by the diploidisation period with descending dysploidy as one of the main chromosomal rearrangements observed during diploidisation [33]. The identification of descending dysploidy as the main mechanism of karyotype evolution in tribe Sobralieae is in accordance with the diploidisation period following polyploid pulses, possibly placed on the ancestral node of Orchidaceae. Since *Brasolia* and *Sobralia* are subject to the same karyotype evolution mechanism and because chromosome number has no phylogenetic signal, it is hard to use the karyotype to differentiate these groups. However, along the descending dysploidy, the elimination of repetitive sequences is another frequent event during the diploidisation period [34], affecting the genome size, which presents a phylogenetic signal.

### 3.3. Genome Size Reconstruction: The Role of Genome Size in Species Evolution

The genera *Brasolia* and *Sobralia* did not differ in genome size, while *Elleanthus* differs from both genera, probably due to the reduced 1C-value of *E. crinipes*. The genome size seems not to have any association with chromosome number variation. For example, *Sobralia* species with 2*n* = 48 vary 2-fold in their genome size (1C-value from 4.34 pg to 8.08 pg), while *Brasolia* species with 2*n* = 46 vary from 4.97 to 6.37 pg, and between the two *Elleanthus* species, both with 2*n* = 50, the genome size varies more than 2-fold (from 2.04 to 4.79 pg). The difference observed in genome size is probably due to differences in repetitive sequences as transposable elements and satellite DNA families [34,35]. The homogenisation and elimination of such redundant sequences are characteristics of the diploidisation process [35] and probably shaped the genomes in alternative ways resulting in genome size differences, even with the same chromosome number [36].

The chromosome number and genome size are essential as reproductive barriers, ensuring the species’ integrity [37]. Here, the most significant difference in genome size, conserving the same chromosome number, was observed in *Elleanthus*. The two studied *Elleanthus* species, besides differing in genome size, also differ in their distribution. While *E. brasiliensis* ranges from French Guiana to southern Brazil and in the Brazilian Atlantic Rain Forest occurring from the sea level until 1000 m, *E. crinipes* is endemic to the highland forest areas in Brazilian Atlantic Rain Forest, occurring only above 800 m [38,39]. Despite the region of sympatry between the two species that present simultaneous blooming and partially share pollinators, no hybrids were described between these two species, with no fruit being formed after manual interspecific crossings [38]. Despite the two species presenting the same chromosome number with similar karyotype morphologies, the two-fold genome size difference should configure as a strong post-zygotic barrier preventing hybridisation, highlighting the importance of genome size as a reproductive barrier and its importance in speciation (reviewed [36]). 

### 3.4. Genus Delimitation in Sobralieae: Future Perspectives

Despite the genera *Brasolia* and *Sobralia* being placed on two groups with high support in the molecular phylogeny [4,5], they present very similar morphology and could not be differentiated based on chromosome number and genome size, as suggested by the ancestral state reconstruction. However, *Elleanthus*, with a distinct morphology, was nested in *Brasolia* [4,5] and present study but also did not show a clear difference in chromosome number and genome size. Our limited sampling of *Elleanthus* prevents further discussion but highlights the importance of future studies in this genus. 

The phylogenetic position of Sobralieae makes studies in this tribe of particular interest to better understand the evolutionary trends in the major subfamily of Orchidaceae. Furthermore, considering that Epidendroideae represents ca. 80% of orchids, better comprehending the diversification of this clade configures as one of the main objectives in orchid diversification and evolution studies [30,40] since such subfamily reflects the diversity of the whole family, and Sobralieae, placed at the base of subfamily Epidendroideae, is of particular interest. 

## 4. Materials and Methods

### 4.1. Plant Material

Plant material for cytological and molecular studies was cultivated in two living orchids collections: of the University of Gdansk, Poland and Federal University of ABC (UFABC). All the experiments were performed in accordance with relevant guidelines and regulations.

### 4.2. Phylogenetic Analysis

For phylogenetic reconstruction we sampled 57 individuals from 53 species distributed in four genera: 11 *Brasolia*, four *Elleanthus*, 34 *Sobralia*, and four *Neottia* species (outgroup) (Appendix A). We sampled species favoring those with available material for cytogenetic data (chromosome number and genome size), but that also represent all groups detected in previous studies [4,5].

Genomic DNA extraction was performed in two ways, both AX Sherlock kit (A&A Biotechnology, Gdansk, Poland) and CTAB 2× protocol [41] excluding β-mercaptoethanol were used. For both protocols, a fragment of plant tissue (~30 mg) and a ceramic sphere were placed in FastPrep tubes and the samples were homogenized. The final pellet was resuspended in 50 µL TE buffer. Two sequences were amplified: nrITS and *mat*K. The nrITS sequences were amplified using two different pairs of primers: 101F and 102R [42] and 17SE and 26SE [43], while, for the *mat*K gene, the primers 19F [44] and 1326R [45] were used for all samples. The PCRs, for both of the markers, were performed in a total volume of 25 µL, containing 1 µL temple DNA (~10–100 ng), 0.5 µL of 10 µM of each primer, 1.0 µL dNTP mix (0.2 mM of each dNTP), 2.5 µL 10× Hybrid buffer contains 15 mM MgCl_2_, 0.5 µL Hybrid DNA Polymerase (2U/µL) and water. The reaction parameters for the nrITS were: 94 °C, 4 min; 30× (94 °C, 45 s; 52 °C, 45 s; 72 °C, 1 min); 72 °C, 7 min. For the *mat*K fragment, we set the following parameters: 95 °C, 3 min; 33× (94 °C, 45 s; 52 °C, 45 s; 72 °C, 2 min 30 s); 72 °C, 7 min. The products of PCR reaction were cleanup using Wizard SV Gel and a PCR Clean Up System (Promega, Madison, WI, USA), following the manufacturer’s protocol. Purified PCR products were sequenced by Macrogen (Seoul, South Korea), using the same primers mentioned above. DNA sequence chromatograms were examined/edited in FinchTV software (https://finchtv.software.informer.com/1.4/ accessed on 12 October 2021). 

The phylogenetic reconstruction was based on the two obtained matrices, from nrITS and *mat*K, aligned in MAFFT v. 7.453 software using progressive alignment and iterative refinement methods [46]. The aligned sequences were trimmed using BioEdit v. 7.2 [47] and concatenated using Mesquite v 3.61 [48]. The generated matrix was analyzed under Maximum Likelihood (ML) and Bayesian Inference (BI). 

The ML analysis was performed using IQ-TREE2 [49], searching for the best-scoring ML tree under the following nucleotide substitution models: TNe+G4 to nrITS and K3Pu + F + G4 to matK, both chosen by ModelFinder applying the Bayesian Information Criterion (BIC) [50]. We used 100 bootstrapping to assess branch supports. The BI was performed using MrBayes v. 3.2.7a [51] with GTR+I+G as the selected evolutive model for both partitions. Four independent runs, one hot and three cold chains, were started from different random trees to ensure that individual runs had converged to the same result. We used 20 million generations per chain with a sampling frequency of 2000. Split frequencies below 0.01 were used to check for convergence and the effective sample size (ESS) for each run was checked in the Tracer v. 1.7.1 [52]. In this case, 25% of trees were excluded as burn-in. Saved trees were summarized in a majority rule consensus tree created with nodal confidence assessed by posterior probabilities (PP), which were strongly supported when equal to or higher than 0.95 [53,54]. The ML and BI trees were edited with FigTree v.1.4.4 (http://tree.bio.ed.ac.uk/software/figtree/ accessed on 12 October 2021) and Adobe Photoshop CS5 (Adobe Systems, Inc., San Jose, CA, USA). Alignments and phylogenetic trees are available from TreeBase (http://treebase.org, submission number 28791, accessed on 12 October 2021). 

### 4.3. Cytogenetics Analysis

For chromosome counting, between one to four specimens per species were analyzed. For each specimen, at least three slides were analyzed, with, at least, five metaphase plates each. Fresh root tips were pretreated with 0.002 M 8-hydroxyquinoline at 4 °C or 10 °C for 12 h. The material was then fixed in a mixture of 99.8% ethanol and glacial acetic acid (3:1, *v*:*v*) and stained according to standard methods with aceto-orcein [55], Feulgen’s stain [56] or DAPI (1 µg mL^−1^) for 30 min. Chromosomes in late prophase and/or metaphase were analyzed using a Nikon Eclipse E800 epifluorescence microscope equipped with differential interference contrast (DIC) optics and a Nikon DS-5Mc CCD camera (PRECOPTIC Co., Warszawa, Poland) or with a fully automated upright fluorescent microscope (Leica DM6000 B, KAWA.SKA Sp. z.o.o., Zalesie Górne, Poland). Photomicrographs were arranged and analyzed in Adobe Photoshop (Elements 11 and CS6; Adobe System Inc., San Jose, CA, USA). 

The genome size estimation was measured using fresh and young leaves of the selected *Brasolia*, *Elleanthus*, and *Sobralia* species. Depending on plant material availability, between one to three specimens per species were analyzed (Table 1), and each specimen was analyzed in triplicate (Table 1). Plant material was prepared as previously described [57], using 1 mL of Tris-MgCl_2_ nucleus isolation buffer [58] supplemented with 1% (*w*/*v*) polyvinylpyrrolidone (PVP-10) for *Brasolia* and *Sobralia* and using Ebihara buffer for *Elleanthus* [59]. The buffers were supplemented with ribonuclease A (50 mg mL^−1^) and propidium iodide (PI, 50 mg mL^−1^). For 25 accessions of *Brasolia* and *Sobralia*, *Secale cereale* cv. Dankowskie (2C = 16.19 pg; [60]) was used as an internal standard, whereas *Pisum sativum* cv. Set (2C = 9.11 pg; [61]) was used for *S. luerorum* accessions and *Vicia faba* cv. Inovec (2C = 26.90 pg; [60]) was used for *Elleanthus* species, avoiding sample and standard peaks overlapping. Genome size was estimated using a CyFlow SL Green (Partec GmbH, Münster, Germany) flow cytometer and a FACS Canto II cytometer (Becton Dickinson, San Jose, CA, USA), kindly made available by the Microbiology and Immunology Department of IBB-UNESP/Botucatu, Brazil, both equipped with a high-grade solid-state laser with green light emission at 532 nm. The 2C DNA content was measured in at least 5000 events using linear amplification. Histograms were evaluated using a FloMax program (Partec GmbH, Münster, Germany) and Flowing Software 2.5.1 (http://www.flowingsoftware.com/ accessed on 12 October 2021). The coefficient of variation (CV) of the G0/G1 peak of the studied species ranged between 2.84 and 5.00%. The nuclear DNA content of each species was calculated using the linear relationship between the ratio of the target species and the internal standard 2C peak positions on the histogram of fluorescence intensities. The 2C DNA contents (pg) were expressed in Mbp using the equation 1 pg = 978 Mbp [62]. The results of FCM estimation were analyzed using a one-way analysis of variance (ANOVA) following by Duncan’s test (*p* < 0.05) in R [63].

### 4.4. Chromosome Number and Genome Size Evolution

The phylogeny reconstructed here was pruned, removing species without information about chromosome number and genome size. The pruning used the function drop.tip from ‘ape’ package [64] in R [63]. The pruned phylogenetic tree was used to reconstruct the ancestral state for chromosome number, using ChromEvol v.2 [65,66] with posterior Adequacy test (Rice and Mayrose 2021) and genome size, using Trace Character History in Mesquite v 3.61 [60]. 

ChromEvol uses likelihood-based methods to evaluate the chromosome number changes along the phylogenetic branches, choosing the model that better performs among the eight available models. The inference considers four parameters: polyploidy (rate ρ), demi-duplication (i.e., a fusion of gametes of different ploidy, with rate μ), and dysploidy, which could be ascending (rate λ) or descending (rate δ). These parameters are combined and tested under two scenarios: constant and linear models. While the former estimates the rate of changes independently of the current chromosome number, the last considers the possibility of chromosome number change depending on the current chromosome number. Both model sets include a null model that assumes no polyploidization events. So, ultimately, the eight models (four constant models and four linear) were tested, allowing us to infer the main mechanism in chromosome number changes and the ancestral haploid number on the nodes of the phylogenetic tree. The best-fitted model was selected using the AIC and, considering the best-fitted model, we re-ran the analysis using 10,000 simulations. The difference among AIC values below two (ΔAIC < 2) were considered not distinguishable [67,68]. The model adequacy test [69] was performed online (http://chromevol.tau.ac.il/ accessed on 12 October 2021) to evaluate the selected model’s capability in generating the observed chromosome numbers. 

Given the overall effect of phylogeny on a variation of chromosome number and genome size, we estimated the phylogenetic signal (i.e., the “tendency for related species to resemble each other more than they resemble species drawn at random from the tree”; [70,71] using Pagel’s lambda (λ; [72]). The index was estimated using phylosig from ‘phytools’ package [73] in R [63].

## Figures and Tables

**Figure 1 ijms-23-03948-f001:**
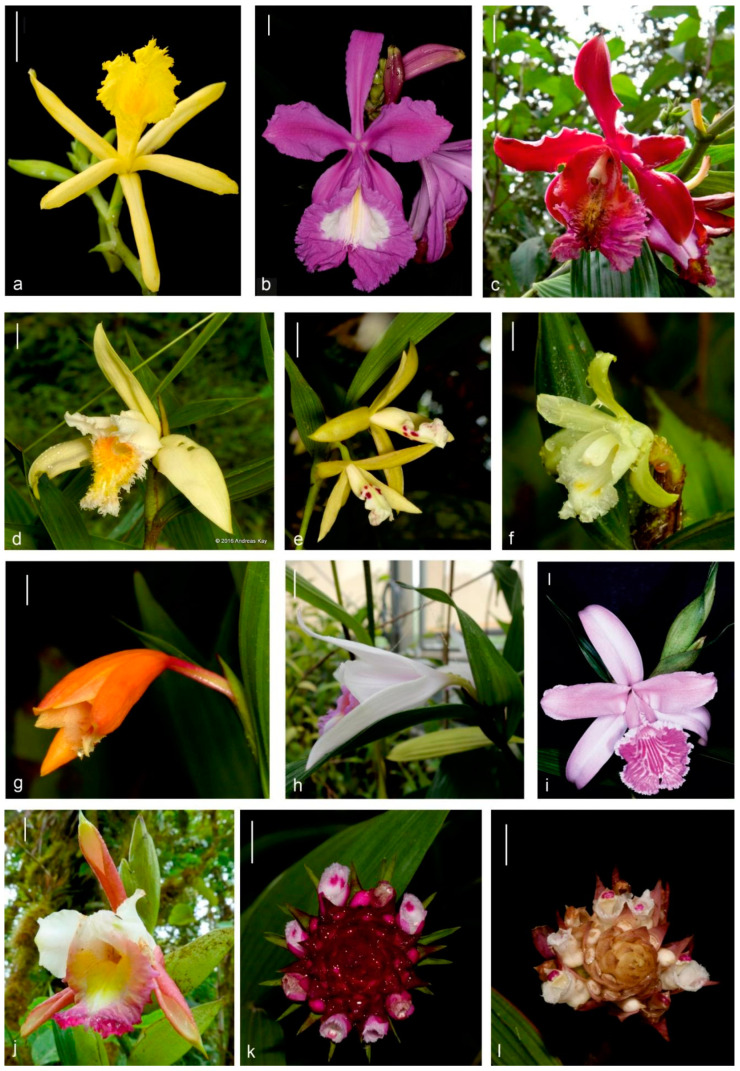
Diversity in *Brasolia*, *Sobralia* and *Elleanthus* flowers. (**a**) *B. flava* (Eric Hunt); (**b**) *B. portillae* (Lourens Grobler); (**c**) *B. dichotoma* (Marta Kolanowska); (**d**) *S. fimbriata* (Andreas Kay); (**e**) *S. lancea* (Andreas Kay); (**f**) *S. bimaculata* (Andreas Kay); (**g**) *S. crocea* (Andreas Kay); (**h**) *S. decora* (Przemyslaw Baranow); (**i**) *S. rosea* (K-Eric Hunt); (**j**) *S. gloriosa* (Andreas Kay); (**k**) *E. brasiliensis* (Ana Paula Moraes); (**l**) *E. crinipes* (Carlos Nunes). Scale bars = 10 mm.

**Figure 2 ijms-23-03948-f002:**
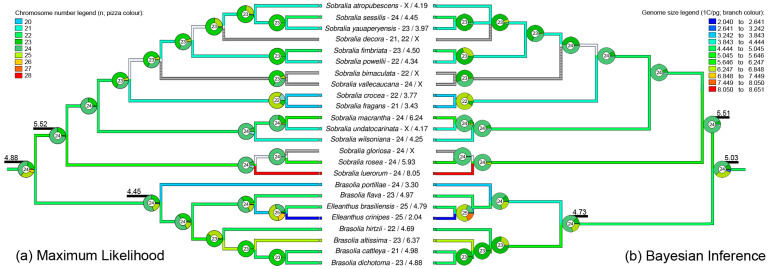
Maximum likelihood (**a**) and Bayesian Inference (**b**) pruned phylogenies showing the ancestral reconstruction of base chromosome number on nodes and genome size (1C-value) on branches. The ancestral genome size is indicated on the node for *Brasolia*, *Sobralia*, and the ancestral of both genera. The colour legend for the base chromosome number is on the right, and for genome size, on the left.

**Figure 3 ijms-23-03948-f003:**
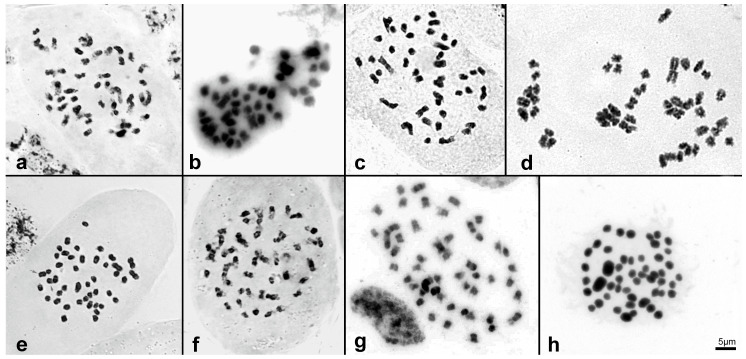
Chromosome complement of *Brasolia* and *Elleanthus* species. (**a**) *B. altissima* (2*n* = 46), (**b**) *B. cattleya* (2*n* = 42), (**c**) *B. dichotoma* (2*n* = 46), (**d**) *B. flava* (2*n* = 46), (**e**) *B. hirtzii* (2*n* = 44), (**f**) *B. portillae* (2*n* = 48), (**g**) *E. brasiliensis* (2*n* = 50), (**h**) *E. crinipes* (2*n* = 50). Bar in h correspond to 5 µm.

**Figure 4 ijms-23-03948-f004:**
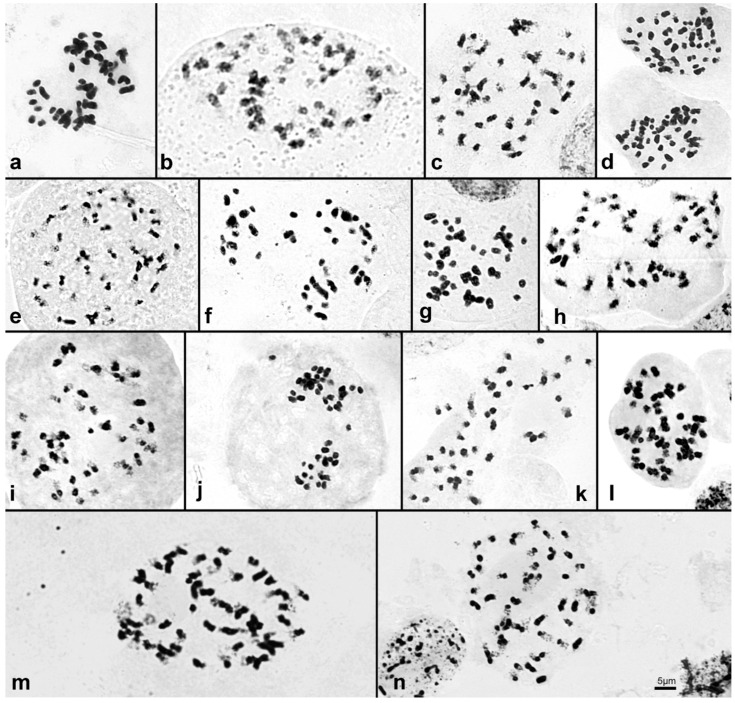
Chromosome complement of *Sobralia* species. (**a**) *S. bimaculata* (2*n* = 44), (**b**) *S. decora* (2*n* = 42), (**c)** *S. decora* (2*n* = 44), (**d**) *S. fimbriata* (2*n* = 46), (**e**) *S. macrantha* (2*n* = 48), (**f**) *S. powelli* (2*n* = 44), (**g**) *S. vallecaucana* (2*n* = 48), (**h**) *S. wilsoniana* (2*n* = 48), (**i**) *S. yauaperyensis* (2*n* = 46), (**j**) *S. crocea* (2*n* = 44), (**k**) *S. fragrans* (2*n* = 42), (**l**) *S. gloriosa* (2*n* = 48), (**m**) *S. luerorum* (2*n* = 48), (**n**) *S. rosea* (2*n* = 48). Bar in h correspond to 5 µm.

**Figure 5 ijms-23-03948-f005:**
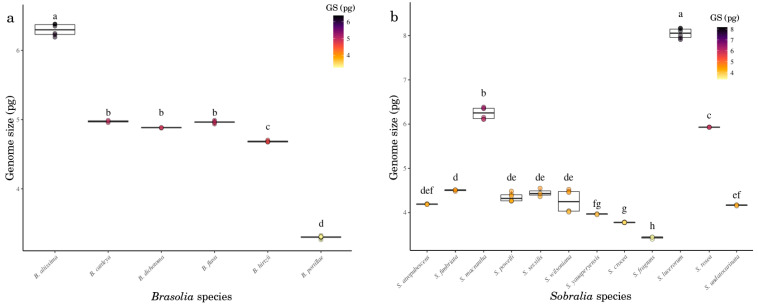
*Brasolia* (**a**) and *Sobralia* (**b**) genome sizes (1C-value). The 1C-value indicated by the same letter are not significantly different at *p* ≤ 0.05 according to the Duncan’s multiple range test.

**Table 1 ijms-23-03948-t001:** Chromosome number and genome size in Sobralieae. The list of *Brasolia*, *Elleanthus*, and *Sobralia* species analyzed here and published on the literature is presented. For each species, the determined chromosome number is presented followed by its figure in text or reference on the literature. For each species analyzed here, the number of analyzed specimens in FCM is indicated in brackets. The genome size is presented in pg, followed by its standard deviation (SD), and Mega base-pairs (Mbp). The genome size (the 1C-value) is given in pg ± SD (Standard Deviation) and Mbp (Mega base-pairs). For each 1C-value, the sample Coefficient of Variation (CV) is presented and the genome size is classified according Soltis et al. [16].

No.	Species [n. FCM Analyzed Specimens]	Chromosome Number	Figure/Reference	1C DNA Content		CV Sample %	Genome Size Category
				pg ± SD	Mbp		
	** *Brasolia* **						
01	*B. altissima* [2]	46	Figure 3a	6.37 ± 0.075	6157	3.65	intermediate
02	*B. cattleya* [1]	42	Figure 3b	4.98 ± 0.014	4876	3.95	intermediate
03	*B. dichotoma* [1]	46	Figure 3c	4.88 ± 0.005	4773	3.81	intermediate
04	*B. flava* [2]	46	Figure 3d	4.97 ± 0.028	4854	4.14	intermediate
05	*B. hirtzii* [2]	44	Figure 3e	4.69 ± 0.011	4582	3.76	intermediate
06	*B. portillae* [3]	48	Figure 3f	3.30 ± 0.016	3228	4.75	small
	** *Elleanthus* **						
07	*E. brasiliensis* [3]	50	Figure 3g/FG10 [8]	4.79 ± 0.075	4687	2.84	intermediate
08	*E. crinipes* [3]	50	Figure 3h	2.04 ± 0.078	1995	3.11	small
	** *Sobralia* **						
	**Sect. *Sobralia***						
09	*S. atropubescens* [1]	-	-	4.19 ± 0.010	4098	5.08	intermediate
10	*S. bimaculata*	44	Figure 3a	-	-	-	-
11	*S. decora*	42, 44	Figure 3b,c	-	-	-	-
12	*S. fimbriata* [1]	46	Figure 3d	4.50 ± 0.019	4401	4.37	intermediate
13	*S. liliastrum*	48	FG10 [8]	-	-	-	-
14	*S. macrantha* [2]	48	Figure 3e	6.24 ± 0.121	6108	4.15	intermediate
15	*S. powelli* [2]	44	Figure 3f	4.34 ± 0.085	4248	4.45	intermediate
16	*S. sessilis* [1]	48	FG10 [8]	4.45 ± 0.095	4347	4.47	intermediate
17	*S. vallecaucana*	48	Figure 3g	-	-	-	-
18	*S. wilsoniana* [2]	48	Figure 3h	4.25 ± 0.025	4163	4.95	intermediate
19	*S. yauaperyensis* [1]	46	Figure 4i	3.97 ± 0.012	3878	4.52	intermediate
	**Sect. *Intermediae***						
20	*S. crocea* [2]	44	Figure 4j	3.77 ± 0.005	3695	5.00	intermediate
21	*S. fragrans* [1]	42	Figure 4k	3.43 ± 0.025	3355	4.52	small
	**Sect. *Racemosae***						
22	*S. gloriosa*	48	Figure 4l	-	-	-	-
23	*S. luerorum* [2]	48	Figure 4m	8.05 ± 0.103	7872	4.14	intermediate
24	*S. rosea* [1]	48	Figure 4n	5.93 ± 0.006	5800	3.48	intermediate
	**Complex *undatocarinata***						
25	*S. undatocarinata* [1]	-	-	4.17 ± 0.013	4074	4.45	intermediate

**Table 2 ijms-23-03948-t002:** ChromEvol model results to infer the ancestral base chromosome number and the chromosome number changes in the tribe Sobraliinae. The analysis was carried out for Maximum Likelihood and Bayesian Inference phylogenetic trees. For each tested ChromEvol model for each phylogenetic method, the Log-likelihood and AIC values are presented. The lowest AIC in each phylogenetic tree is indicated in bold and * indicates alternative model that does not differ from the lowest AIC.

MODELS	Maximum Likelihood		Bayesian Inference	
	Log-Likelihood	AIC	Log-Likelihood	AIC
Constante Rate	−51.76	109.5 *	−47.99	102 *
Constante Rate with demi-polyploid	−51.76	109.5 *	−47.99	102 *
Constante Rate with demi-polyploidy *est*	−51.76	111.5	−47.99	104
Constante Rate without polyploid (null model)	−51.76	**107.5**	−47.99	**99.99**
Linear rate	−50.92	111.8	−47.71	105.4
Linear rate with demi-polyploid	−50.92	111.8	−47.71	105.4
Linear rate with demi-polyploid *est*	−50.92	113.8	−47.69	107.4
Linear rate without polyploid (null model)	−50.92	109.8	−47.71	103.4

**Table 3 ijms-23-03948-t003:** Pagel’s lambda and Blomberg’s K phylogenetic signals for chromosome number and genome size under Maximum Likelihood and Bayesian Inference.

Index/Phylogenetic Reconstruction	Chromosome Number		Genome Size	
	Maximum Likelihood	Bayesian Inference	Maximum Likelihood	Bayesian Inference
Pagel’s λ	0.891 (*p* = 0.0003)	0.885 (*p* = 0.0002)	0.9875 (*p* = 4.83 × 10^−7^)	0.9897 (*p* = 3.53 × 10^−7^)
Blomberg’s K	0.276 (*p* = 0.003)	0.523 (*p* = 0.001)	1.00 (*p* = 0.001)	2.20 (*p* = 0.001)

## Data Availability

Alignments and phylogenetic trees are available from TreeBase (http://treebase.org, submission number 28791, accessed on 12 October 2021).

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
