# Peer review of "Chromosome Number and Genome Size Evolution in *Brasolia* and *Sobralia* (Sobralieae, Orchidaceae)"

_ijms, 2022, doi:10.3390/ijms23073948_

Round 1
Reviewer 1 Report
The paper is well written, nicely presented with interesting results and discussion. I recommend the publication of the paper in International Journal of Molecular Sciences after minor revisions:
Line 4-5: superscript signs need revision
Line 40: how tall is “very tall”? I suggest including a measurement.
Figure 1: I suggest including a scale bar. The distance between each flower picture are very unequal, please fix it.
Lines 104-113: genera and species names should be in italic.
Line 115: remove italic from the name Figure 2
Line 169: remove italic from the name “and”.
Lines 213-229: genera and species names should be in italic.
Discussion and Material and Methods: please, genera and species names should be in italic.
Line 256: change polyploid to polyploidy
Line 258: change polyploid to polyploidy
It is polyploid or polyploidy pulse?
Line 261: change 5Mya to 5 Mya
Line 265: fix (Mya; [27-30).
Line 341: “Extraction, amplification and sequencing” is a subsection? I suggest organizing the methodology into subsections.
Line 383: please, note that the degrees Celsius symbol needs to be standardized throughout the methodology.

Author Response
Dear Reviewer,
thank you for your review and comments. We accepted an commented all your sugestions, which improved our manuscript. Please, find the detailed answer to each reviewer question below.
Regards,
Joanna Rojek,
on behalf of authors
Answer to Rev #1
R#1: Line 4-5: superscript signs need revision
Authors: It has been done. See lines 4-17.
R#1: Line 40: how tall is “very tall”? I suggest to include a measurement.
Authors: We added an example of this measurement on lines 41-42: “[..] usually tall, or very tall (for example, stems of Sobralia luerorum Dodson reach up to 3.5 m height) [..]”.
R#1: Figure 1: I suggest to include a scale bar. The distance between each flower picture are very unequal, please fix it.
Authors: It has been fixed and scale bar corresponding to 10mm were added in each photo.
R#1: Lines 104-113; Lines 213-229; Discussion and Material and Methods: genera and species names should be in italic.
Authors: Thank you for this correction. It has been fixed in all parts suggested and we double check the genera and species name throughout the text.
R#1: Line 115: remove italic from the name Figure 2.
Authors: Thank you for this correction. This paragraph needed to be updated.
R#1: Line 169: remove italic from the name “and”.
Authors: Thank you for this correction. This paragraph needed to be updated.
R#1: Line 256 and 258: change polyploid to polyploidy
Authors: It has been fixed. See lines 273 and 275.
R#1: It is polyploid or polyploidy pulse?
Authors: In line 275 we mean “Polyploidy pulses” which shoould have occurred in specific geological periods. In line 279, we prefer to keep it as polyploidy and not polyploidy pulses. For the sake of clarity we change to “polyploidy events”.
R#1: Line 261: change 5Mya to 5 Mya
Authors: It has been fixed. See line 278.
R#1: Line 265: fix (Mya; [27-30).
Authors: It has been fixed. See line 282.
R#1: Line 341: “Extraction, amplification and sequencing” is a subsection? I suggest organizing the methodology into subsections.
Authors: It has been fixed. Extraction, amplification and sequencing is part of the Phylogenetic analysis subsection. We modified as suggested, as can be seen from line 359 to 399.
R#1: Line 383: please, note that the degrees Celsius symbol needs to be standardized throughout the methodology.
Authors:: It has been fixed as can be seen at lines 370-373 and 404.

Reviewer 2 Report
It is interesting manuscript and the results are promissing for the future studies. It is obvious that you have used timid sample quantity (Elleanthus), but is a real part of your work. The methods are appropriate and well executed.
I have just some notes for you to correct:
Abstract: Despite the clear circumscription of tribe Sobralieae, its internal relationships are still dubious with the recent delimited genus Brasolia, based on previous Sobralia species, being assumed as paraphyletic, with Elleanthus nested on it.
Comment: Here is something missing. You tried to say something, but not explain what. Please change/add something that this phrase can be more clear.
Line 23: Aiming to evaluate the potential of cytogenetic data for Sobralia-Brasolia-Elleanthus genera delimitation, here we present chromosome number and genome size data for 22 and 20 species, respectively.
Comment: If you evaluate 3 genera why you present 2 groups of species (22 and 20)? Need to be clear what this "respectively" means. You have evaluated the chromosome number and genome size per genera or per species or per plant/s?
Line 77: The family Orchidaceae Juss. is known by its large variation .....
Comment: Only here you are mentioning the family. Why? You need to start this Introduction (or Abstract) explaining where your genus belongs to. No infrmation about it also in the Abstract.
In the MM part you have not mentioned how many plants per species you were using. Why? It should be clear in the text. What are individuals? Explain.
LIne 341: Extraction, amplification and sequencing.
Comment: Is this subtitle? Use numbers or just clean it.
Author Response
Dear Reviewer,
thank you for your review and comments. We accepted an commented all your suggestions, which improved our manuscript. Please, find the detailed answer to each question below.
Regards,
Joanna Rojek,
on behalf of authors
Answer to Rev #2
Abstract: Despite the clear circumscription of tribe Sobralieae, its internal relationships are still dubious with the recent delimited genus Brasolia, based on previous Sobralia species, being assumed as paraphyletic, with Elleanthus nested on it.
R#2: Here is something missing. You tried to say something, but not explain what. Please change/add something that this phrase can be more clear.
Authors: Thank you for this comment. Here we mean that despite the tribe Sobralieae being considered monophyletic and clearly circumscribed, some internal relationships are challeging. For example, the recent genus Brasolia is paraphyletic with Elleanthus nested on it. We change the sentence to make it clear: “Despite the clear circumscription of tribe Sobralieae (Orchidaceae), its internal relationships are still dubious. The recently delimited genus Brasolia, based on previous Sobralia species, is now assumed to be paraphyletic, with a third genus, Elleanthus, nested in it.” See lines 18-21.
Line 23: Aiming to evaluate the potential of cytogenetic data for Sobralia-Brasolia-Elleanthus genera delimitation, here we present chromosome number and genome size data for 22 and 20 species, respectively.
R#2: If you evaluate 3 genera why you present 2 groups of species (22 and 20)? Need to be clear what this "respectively" means. You have evaluated the chromosome number and genome size per genera or per species or per plant/s?
Authors: Thank you for this comment. We have two groups of analysis: chromosome number and genome size and the word “respectively” refers to these two analysis. We corrected the number of new chromosome data (21 species) and update the sentence: “Aiming to evaluate the potential of cytogenetic data for Brasolia-Elleanthus-Sobralia genera delimitation, we present chromosome number and genome size data for 21 and 20 species, respectively and used such data to infer the pattern of karyotype evolution in these genera. “ on lines 24-26.
Line 77: The family Orchidaceae Juss. is known by its large variation .....
R#2: Only here you are mentioning the family. Why? You need to start this Introduction (or Abstract) explaining where your genus belongs to. No infrmation about it also in the Abstract.
Authors: Thank you for this comment. The first reference to the plant family is on the title [Chromosome number and genome size evolution in Brasolia and Sobralia (Sobralieae, Orchidaceae)]. Neverthelass, we included the name of the plant family on line 18 (Abstract): “Despite the clear circumscription of tribe Sobralieae (Orchidaceae), its...” and on line 37 (Introduction): “Since Ruiz and Pavón established the genus Sobralia (Orchidaceae) in 1794 [1]...”.
R#2: In the MM part you have not mentioned how many plants per species you were using. Why? It should be clear in the text. What are individuals? Explain.
Authors: Thank you for this comment. For phylogenetic analysis, please, note that we present the number of individuals per species on Table S1, but on methods, line 354-356: “For phylogenetic reconstruction we sampled 57 individuals from 53 species distributed in four genera: 11 Brasolia, four Elleanthus, 34 Sobralia, and four Neottia species (outgroup) (Table S1).”
For chromosome number, one to four specimens per species were used, this information is now included on line 402-403: “For chromosome counting, between one to four specimens per species were analyzed. For each specimen, at least three slides were analyzed, with, at least,five metaphase plates each.”
Finally, to genome size analysis, the number of individuals analysed per species was presented on Table 1, first colunm, and on methods, lines 413-416: “The genome size estimation was measured using fresh and young leaves of the selected Brasolia, Elleanthus, and Sobralia species. Depending on plant material availability, between one to three specimens per species were analyzed (Table 1), and each specimen was analyzed in triplicate (Table 1).”.
Individuals mean specimens. Each individual is one plant. For the sake of clarity we opt to change individual per specimen.
Line 341: Extraction, amplification and sequencing.
R#2’: Is this subtitle? Use numbers or just clean it.
Authors: Thank you for this comment. We re-organized the section Material and Methods in order to make clear the subsections and subtitles.

Reviewer 3 Report
The manuscript of Baranow et al. tries to elucidate the karyotype evolution within Sobralia-Brasolia-Elleanthus by combining phylogenetic data with chromosome countings and genome size measurements of 22 and 20 species, respectively. However, although variations are found in the chromosome number as well as in the genome size the karyotype evoltution in the tribe Sobralieae remains largely unsolved. Without applying additional methods, like e.g. FISH with rDNA or tandem repeat probes precise karyotype rearrangements cannot be solved. Nevertheless, the paper includes some interesting data worth being published after some modifications.
The chromosome numbers are a bit puzzling to me. Exemplarily I tried to count the chromosomes on some of the presented metaphase images. And in some of the cases I ended up with different numbers as postulated, e.g. B. hirtzii: 44 instead of 42, S. rosea:46 instead of 48, S. wilsoniana: 50/52 instead of 48 and so on. However, I agree that counting of chromosomes especially in species with high numbers of small chromosomes is difficult. To convince the readers about the proposed chromosome numbers I suggest to add either little numbers (would probably be too crowded) or dots on or next to the individual chromosomes to indicate what was counted as a chromosome. Furthermore I miss information on how many metaphases were counted per species. Similarly, also the information about number of flow cytometric measurements per individual is missing.
I did not get why certain species were only investigated for the chromosome number while others are only investigated for the genome size. I am especially wondering about the cases where chromosomes could be counted but genome size data are missing. The availability of plant tissue could not be the reason.
The headlines in the manuscript are not informative at all, especially in the results section they simply reflect the used method. This should be changed.
Throughout the manuscript the species and genus names are either in Italics or not. This should be unified.
Minor points:
Line 15: 4Federal University of ABC (UFABC), Center of Natural and Human Science, São Bernardo do Campo, São Paulo, Brazil - Why is this affiliation listed here? None of the authors is listed at 4.
Line 19:”the recent delimited genus” -> ”the recently delimited genus”
Line 20: “nested on” or “nested in”? I would prefer the latter variant.
Line 48: “flowers positioned in a terminal and condensed inflorescences” -> “flowers positioned in terminal and condensed inflorescences”
Line 58: “relations, Sobralia section Sobralia was” -> ““relations, the section Sobralia was”
Line 70: “significantly help understand the“ -> „“significantly help to understand the“
Line 71: “only three chromosomes counts” -> “only three chromosome counts”
Line 104: “methodologies (ML and BI) returned the same” -> “methodologies (ML and BI) revealed the same”
Line 155: “and internal variance among individuals are also presented” – this information is not shown in the table
Line 171: “For each ested ChromEvol model” -> “For each tested ChromEvol model”
Line 186: “could not differ Sobralia and Brasolia” -> “could not differentiate Sobralia and Brasolia”
Line 187: “size suggested an higher ancestral 1C-value for Sobralia presented a higher genome size (1C=5.5pg) compared to Brasolia” -> “size suggested an higher ancestral 1C-value for Sobralia (1C=5.5pg) compared to Brasolia”
Line 190: “For example, the three species S. macrantha (1C=6.24pg), S. rosea (1C=5.93pg), and S. luerorum (1C=8.05pg), even presenting different 1C-values (Fig. S1) present the same chromosome number 2n=48 (n=24; Fig. 2).” -> “For example, the three species S. macrantha (1C=6.24pg), S. rosea (1C=5.93pg), and S. luerorum (1C=8.05pg), vary severely in the 1C-values (Fig. S1) although they present the same chromosome number 2n=48 (n=24; Fig. 2).”
Line 193: What means in one accession? Were several accessions investigated?
Line 194: check the genome size of B. dichotoma
Line 195: check the genome size of S. crocea, the table shoes 3.77
Line 227: “distinct (as seen at Fig. 1A-C with K and L)and the Elleanthus” -> “distinct (as seen at Fig. 1A-C with K and L) and the Elleanthus”
Line 256: “The polyploid is a central” -> “Polyploidization is a central”
Line 278: ” possible placed“ -> „possibly placed”
Line 279: “Since the three studies genera” -> “Since the three studied genera”
Line 281: “hard to differ the karyotype among these groups” -> “hard to differentiate the karyotypes among these groups”
Line 294: “more tha 2-fold“ -> „more than 2-fold”
Line 296: “homogeneisation“ -> „homogenisation“
Line 297: „characteristics from diploidisation” -> “characteristics of the diploidisation”
Line 304: “in endemic” -> “is endemic”
Line 319:”our timid sampling” -> “our limited sampling”
Line 320: “ highlight the” -> “highlights the”
Line 321: “ makes the studies” -> “makes studies”
Line 322: “subfamily in Orchidaceae” -> “subfamily of Orchidaceae”
Line 407: “depends on plants material” -> “depending on plant material”
Fig. 2: To better see potential correlations between the genome size and chromosome numbers I would suggest to use on both cases the same color code. Otherwise it is quite difficult to read when in one case red color indicates a large genome size while in case of chromosomes it indicates a low number.
Author Response
Dear Reviewer,
thank you for your review and comments. We accepted an commented all your suggestions, which improved our manuscript. Please, find the detailed answer to each question below.
Regards,
Joanna Rojek,
on behalf of authors
Answer to Rev #3
R#3: The manuscript of Baranow et al. tries to elucidate the karyotype evolution within Sobralia-Brasolia-Elleanthus by combining phylogenetic data with chromosome countings and genome size measurements of 22 and 20 species, respectively. However, although variations are found in the chromosome number as well as in the genome size the karyotype evoltution in the tribe Sobralieae remains largely unsolved. Without applying additional methods, like e.g. FISH with rDNA or tandem repeat probes precise karyotype rearrangements cannot be solved. Nevertheless, the paper includes some interesting data worth being published after some modifications.
Authors: Thank you for your comments. We agree that based on your present results, further discussions about the mechanisms of chromosome rearrangments will need to apply bioinformatic analysis to isolated repetitive DNA sequences and in situ hybridisation techniques to located such sequences. We added a comment about this on line 268-271: “However, based on the homogenous chromosome morphology presented in the species analyzed here, it was impossible to infer the fission-fusion points or chromosome loss, causing the chromosome number reduction, making mandatory the use of in situ hybridization techniques.”
R#3: The chromosome numbers are a bit puzzling to me. Exemplarily I tried to count the chromosomes on some of the presented metaphase images. And in some of the cases I ended up with different numbers as postulated, e.g. B. hirtzii: 44 instead of 42, S. rosea: 46 instead of 48, S. wilsoniana: 50/52 instead of 48 and so on. However, I agree that counting of chromosomes especially in species with high numbers of small chromosomes is difficult. To convince the readers about the proposed chromosome numbers I suggest to add either little numbers (would probably be too crowded) or dots on or next to the individual chromosomes to indicate what was counted as a chromosome.
Authors: Thank you for this comment. As you wrote, counting small and numerous chromosomes is challenging. Precisely because of it, the chromosome number for each species was checked in three slides with at least five metaphases plate in each. All metaphases were analysed independently by four authors. At this opportunity, three of us (JR, JB, and APM) rechecked the majority of metaphases, considering your corrections. We change some cells from Fig 3 and 4 to make it easier to convince the readers, specifically the cells from species S. wilsoniana, S. fragrans, S. luerorum, and S. rosea. We also identified some equivocal chromosome numbers. In this sense, we update the chromosome counting to S. decora (with two cytotypes, 2n=42 and 44), S. fimbriata (2n=46), and B. hirtzii (2n=44). Besides, we followed your suggestion and included, as supplementary material, a figure with all chromosomes identified by numbers.
R#3: Furthermore I miss information on how many metaphases were counted per species. Similarly, also the information about number of flow cytometric measurements per individual is missing.
Authors: Thank you for this suggestion. We added information about number of species, specimens per species, slides per specimens and cells per slide in ‘material and methods’ sections, lines 401-403: “For chromosome counting, between one to four specimens per species were analyzed. For each specimen, at least three slides were analyzed, with, at least,five metaphase plates each.”
Regarding flow cytometry, this information was already included on the previous version. It can be found now on Lines 414-416: “Depending on plant material availability, between one to three specimens per species were analyzed (Table 1), and each specimen was analyzed in triplicate (Table 1).”. The information previously placed on Table 1 is maintened.
R#3: I did not get why certain species were only investigated for the chromosome number while others are only investigated for the genome size. I am especially wondering about the cases where chromosomes could be counted but genome size data are missing. The availability of plant tissue could not be the reason.
Authors: Thank you for this question. For five species we could get only chromosome number (S. bimaculata, and S. decora S. vallecaucana, S. gloriosa) or only genome size (S. atropubescens) data. Cultivate these Neotropical species in Polland was challenging and, unfortunatelly, we lost part of the collection before we could begin the experiment of flow cytometry. Regarding the S. atropubescens, even after many tries, we could not get enough metaphases cells that allowed us to determine the chromosome number.
R#3: The headlines in the manuscript are not informative at all, especially in the results section they simply reflect the used method. This should be changed.
Authors: We modified the headlines to reflect the information placed on each section.
R#3: Throughout the manuscript the species and genus names are either in Italics or not. This should be unified.
Authors: It has been fixed throughout the manuscript.
Minor points:
Line 15: 4Federal University of ABC (UFABC), Center of Natural and Human Science, São Bernardo do Campo, São Paulo, Brazil - Why is this affiliation listed here? None of the authors is listed at 4.
Authors: Thank you for this correction. It is listed because it is the affiliation of Ana Paula Moraes. This affiliation was updated on Page 1, Line 15 to “Laboratory of Cytogenomic and Evolution of Plants, Center of Natural and Human Science, Federal University of ABC (UFABC), São Bernardo do Campo, São Paulo, Brazil; ana.moraes@ufabc.edu.br”
Line 19: ”the recent delimited genus” -> ”the recently delimited genus”
Line 20: “nested on” or “nested in”? I would prefer the latter variant.
Authors: The sentence on Lines 19-20 were updated to “The recently delimited genus Brasolia, based on previous Sobralia species, is now assumed to be paraphyletic, with a third genus, Elleanthus, nested in it.”
Line 48: “flowers positioned in a terminal and condensed inflorescences” -> “flowers positioned in terminal and condensed inflorescences”
Authors: Thank you for this suggestion. We remove the article ‘a’ from the sentence, now on Page 2, Line 49.
Line 58: “relations, Sobralia section Sobralia was” -> ““relations, the section Sobralia was”
Authors: Thank you for this suggestion. We update the sentence, now on Page 2, Line 59.
Line 70: “significantly help understand the“ -> „“significantly help to understand the“
Authors: Thank you for this suggestion. We update the sentence, now on Page 2, Line 71.
Line 71: “only three chromosomes counts” -> “only three chromosome counts”
Authors: Thank you for this suggestion. We update the sentence, now on Line 73
Line 104: “methodologies (ML and BI) returned the same” -> “methodologies (ML and BI) revealed the same”
Authors: Thank you for this suggestion. We update the sentence to “Both applied methodologies (ML and BI) revealed the same general results”, now on Line 107
Line 155: “and internal variance among individuals are also presented” – this information is not shown in the table
Authors: Thank you for this comment. Since the variance among individuals were below 1%, it could be considered negligenciable. We opt to remove this sentence from the Table 1 and, now, we removed from the title of Table 1, Line 163.
Line 171: “For each ested ChromEvol model” -> “For each tested ChromEvol model”
Authors: Thank you for this correction. We correct the word “tested” in the legend of Table 2, now on Line 183
Line 186: “could not differ Sobralia and Brasolia” -> “could not differentiate Sobralia and Brasolia”
Line 187: “size suggested an higher ancestral 1C-value for Sobralia presented a higher genome size (1C=5.5pg) compared to Brasolia” -> “size suggested an higher ancestral 1C-value for Sobralia (1C=5.5pg) compared to Brasolia”
Authors: Thank you for these two corrections. We update the sentence on Line 199-202: “Despite statistics test could not differentiate Brasolia and Sobralia based on genome size (F=0.488; p=0.487), the ancestral reconstruction for genome size suggested a higher ancestral 1C-value for Sobralia (1C=5.5pg) compared to Brasolia (1C=4.45-4.73pg) (Fig. 2).”
Line 190: “For example, the three species S. macrantha (1C=6.24pg), S. rosea (1C=5.93pg), and S. luerorum (1C=8.05pg), even presenting different 1C-values (Fig. S1) present the same chromosome number 2n=48 (n=24; Fig. 2).” -> “For example, the three species S. macrantha (1C=6.24pg), S. rosea (1C=5.93pg), and S. luerorum (1C=8.05pg), vary severely in the 1C-values (Fig. S1) although they present the same chromosome number 2n=48 (n=24; Fig. 2).”
Authors: Thank you for this suggestion. We update this sentence, including S. wilsoniana: “For example, the species S. wilsoniana (1C=4.25pg), S. rosea (1C=5.93pg), S. macrantha (1C=6.24pg), and S. luerorum (1C=8.05pg) strongly vary among their 1C-values (Fig. S1), although they present the same chromosome number 2n=48 (n=24; Fig. 2).” See Line 203-206.
Line 193: What means in one accession? Were several accessions investigated?
Line 194: check the genome size of B. dichotoma
Authors: Thank you for this correction. We do not have different measurements for B. dichotoma so we corrected the sentence to “Inside Brasolia, genome size increase could be observed in B. altissima (1C=6.37pg, 2n=46; see Fig. 2).” See Line 206-207.
Line 195: check the genome size of S. crocea, the table shoes 3.77
Authors: Thank you for this correction. We corrected the 1C-value to 3.77pg on Line 208.
Line 227: “distinct (as seen at Fig. 1A-C with K and L)and the Elleanthus” -> “distinct (as seen at Fig. 1A-C with K and L) and the Elleanthus”
Authors: Thank you for this correction. We corrected the spaces on this sentence on Line 234 and 236.
Line 256: “The polyploid is a central” -> “Polyploidization is a central”
Authors: Thank you for this correction. We updated the sentence to “Polyploidy is a central...” on Line 273.
Line 278: ” possible placed“ -> „possibly placed”
Authors: Thank you for this correction. We change “possible” to “possibly” on Line 297.
Line 279: “Since the three studies genera” -> “Since the three studied genera”
Authors: Thank you for this correction. Since we updated the text, this sentence needed to be deleted.
Line 281: “hard to differ the karyotype among these groups” -> “hard to differentiate the karyotypes among these groups”
Authors: Thank you for this correction. We update the sentence to: “Since Brasolia and Sobralia are subject to the same karyotype evolution mechanism and because chromosome number has no phylogenetic signal, it is hard to use the karyotype to differentiate these groups.” See Line 298-300.
Line 294: “more tha 2-fold“ -> „more than 2-fold”
Authors: Thank you for this correction. We corrected the word “than” on Line 310.
Line 296: “homogeneisation“ -> „homogenisation“
Authors: Thank you for this correction. We corrected the word “homogenisation” on Line 313.
Line 297: „characteristics from diploidisation” -> “characteristics of the diploidisation”
Authors: Thank you for this correction. We change “from” to “ of the” on Line 314.
Line 304: “in endemic” -> “is endemic”
Authors: Thank you for this correction. We corrected the word “is” on Line 322.
Line 319:”our timid sampling” -> “our limited sampling”
Authors: Thank you for this suggestion. We change “timid” by “limited” on Line 337.
Line 320: “ highlight the” -> “highlights the”
Authors: Thank you for this correction. We corrected the word “highlights” on Line 338.
Line 321: “ makes the studies” -> “makes studies”
Authors: Thank you for this correction. We removed the word “the” on Line 340.
Line 322: “subfamily in Orchidaceae” -> “subfamily of Orchidaceae”
Authors: Thank you for this correction. We change “in” to “of” Line 341.
Line 407: “depends on plants material” -> “depending on plant material”
Authors: Thank you for this correction. We modified as required and moved the sentence to the Lines 413-416: “The genome size estimation was measured using fresh and young leaves of the selected Brasolia, Elleanthus, and Sobralia species. Depending on plant material availability, between one to three specimens per species were analyzed (Table 1), and each specimen was analyzed in triplicate (Table 1).”
Fig. 2: To better see potential correlations between the genome size and chromosome numbers I would suggest to use on both cases the same color code. Otherwise it is quite difficult to read when in one case red color indicates a large genome size while in case of chromosomes it indicates a low number.
Authors: Thank you for this suggestion. We edited the image as required.
